# Plant Factory in a Restaurant: Light Quality Effects on the Development, Physiology, and Quality of Three Baby-Leaf Vegetables

**DOI:** 10.3390/plants14020153

**Published:** 2025-01-07

**Authors:** Filippos Bantis, Nikolaos Simos, Athanasios Koukounaras

**Affiliations:** 1Department of Agriculture, University of Western Macedonia, 53100 Florina, Greece; 2School of Agriculture, Aristotle University of Thessaloniki, 54124 Thessaloniki, Greece; simoniko@agro.auth.gr (N.S.); thankou@agro.auth.gr (A.K.)

**Keywords:** lettuce, kale, pak choi, PFAL, vertical farming, OJIP transient, antioxidants

## Abstract

Plant factories with artificial lighting (PFALs) are a notable choice for urban agriculture due to the system’s benefits, where light can be manipulated to enhance the product’s yield and quality. Our objective was to test the effect of light spectra with different red-blue combinations and white light on the growth, physiology, and overall quality of three baby-leaf vegetables (green lettuce, kale, and pak choi) grown in a restaurant’s PFAL. Leaf mass per area was lower under the most blue-containing treatments in all species. The performance indices (PI_abs_ and PI_tot_) of the photosynthetic apparatus were lower under more red light with the exception of PI_abs_ in pak choi. Total soluble solids accumulation was diminished under most of the blue-containing LEDs, while total phenolics and antioxidant activity were induced by red-blue environments rich in blue light. Moreover, chlorophyll and carotenoid accumulation was also enhanced under blue-rich light treatments. Nitrate content was the lowest under monochromatic blue in all species. Finally, the employees were asked about their views on the PFAL within the restaurant’s compounds and they expressed positive opinions. Overall, a light environment including red and blue wavelengths proved beneficial for baby leafy vegetable production in terms of yield and quality.

## 1. Introduction

The International Food and Agriculture Organization (FAO) projects that by 2050, the agricultural sector’s intensive food production will demand more resources to feed the world’s expanding population. With an additional 0.9–5.6 °C increase in average temperature from the last two decades of the 20th century, depending on greenhouse gas emission scenarios, and a 4–22% decrease in annual precipitation, the Mediterranean region is a hotspot for climate change, making the agri-food sector in the region vulnerable [1]. Notably, 21–37% of the world’s annual greenhouse gas emissions are thought to be caused by the present food supply chain, which includes production, transportation, processing, manufacturing, packaging, storage, retail, consumption, losses, and waste [2].

Urban agriculture is characterized by benefits such as the availability of fresh produce with increased storage possibilities. Among urban agriculture systems, indoor vertical farming systems, namely plant factories with artificial lighting (PFAL), are a notable choice by researchers, producers, and stakeholders due to their significant advantages. These are vertical indoor hydroponic farming systems and can be installed in unused buildings (e.g., abandoned warehouses, railway wagons, and old containers) and even operating establishments (e.g., restaurants and supermarkets) while being fully isolated from external environmental conditions [3]. They can also be installed for educational purposes in community centers and schools to introduce locals and students to sustainable vegetable production methods and farming methods while also providing a fresh supply of locally grown vegetables. Utilizing vertical farming systems is a great way to grow vegetables sustainably in densely populated places with a shortage of area for growing, and resource efficiency is crucial. The land-use efficiency of leafy vegetable production in PFALs is significantly higher than that of greenhouses [4]. Additionally, there are far fewer “food miles” and shelf life is expectedly higher due to the shorter time needed for food transportation to the market [5].

Accurate environmental control is a critical parameter that distinguishes PFALs from other farming systems. Light is provided in vertical farming systems exclusively by artificial light sources while light-emitting diodes (LEDs) have displaced traditional light sources (e.g., fluorescent lamps) due to the significant advantages they offer [6]. Light affects plants through its spectrum, intensity, and duration (i.e., photoperiod). Red (600–700 nm) and blue (400–500 nm) wavelengths are considered the main spectral bands that drive photosynthesis and trigger photomorphogenesis. For instance, it was discovered that when blue light levels increased, mustard and kale microgreens became less elongated and accumulated greater amounts of macro- and micronutrients [7]. It is widely acknowledged that species dependency results in varying morphology and nutritional quality under the effect of varying red/blue light ratios. Furthermore, because broad-spectra (or white) light sources enable scouting without sacrificing crop quality or production, they are frequently utilized in place of monochromatic and bichromatic lights. In another study, baby rocket leaves grown under a white light environment with high red content showed the increased expression of genes associated with photosynthetic components, and SUS1 genes involved in accumulating soluble solids, including sugars [8]. In addition, different lighting conditions can affect plant nutrient uptake and nitrogen metabolism [9].

As mentioned above, baby-leaf vegetables are among the crops most widely cultivated in PFALs due to their compact nature, as well as their nutraceutical value, appealing image, and diverse taste, which makes them popular for restaurants, hotels, and domestic consumption [6]. Among various leafy vegetables, including lettuce, members of the Brassicaceae family, such as pak choi and kale, are in increasing demand due to their high nutritional content and unique taste, color, and shape, which can be used to prepare new dishes and garnishes [10].

To maximize land-use efficiency, PFALs are typically characterized by multi-tier structures, where each level has a short vertical distance approximately 50 cm from the next one. To that end, leafy vegetables are among the most suitable crop types for PFAL systems. Our objective was to test the effect of light spectra with different red-blue combinations, as well as white light, on the growth, physiology and overall quality of green lettuce (*Lactuca sativa*, family Asteraceae), kale (*Brassica oleracea* var. *sabellica*, family Brassicaceae), and pak choi (*Brassica oleracea* var. *chinensis*, family Brassicaceae) grown in a restaurant’s PFAL as baby leaves. In addition, the opinions of the staff were also evaluated with respect to the establishment and operation of such a system within the restaurant’s compounds. Contrary to most research studies conducted in growth rooms under fully controlled conditions, our research focuses on the potential baby-leaf cultivation in the busy spaces of a restaurant. Therefore, we did not aim to simulate the cultural conditions of a production unit, but to test the cultivation of leafy vegetables close to the actual consumer’s plate, in the final stage of the “farm-to-fork” chain. An important aim was to test whether the traditionally used light treatments are suitable for such establishments with respect to the obtained product quality, as well as the practicality for employees and the acceptance from customers (secondary aim). This study provided us with the opportunity to evaluate the real-life challenges and benefits during the actual operation of a restaurant. To our knowledge, this is the first article that provides research findings of a plant factory established in a restaurant instead of a fully controlled growth room.

## 2. Results

The light treatments are labeled as follows. R: monochromatic red; B: monochromatic blue; R80-B20: 80% red and 20% blue; R20-B80: 20% red and 80% blue; and W: white.

Plants from all species grew normally with no visible morphological or physiological disorders. In lettuce, leaf mass per area (LMA) was significantly greater under R compared to the rest of the treatments, followed by R80-B20 and W, then R20-B80, while B showed the lowest values (162% lower than R). Quite similarly, pak choi exhibited greater LMA under R, R80-B20, and W compared to R20-B80 and B, with the latter also showing the lowest values (91–109% lower than W, R80-B20, and R). Conversely, kale’s LMA was greater under R20-B80 and W compared to the rest of the treatments (Figure 1A).

PI_abs_ and PI_tot_ in lettuce showed significantly greater values in R20-B80 compared to R and R80-B20. In kale, B imposed greater PI_abs_ compared to R, R80-B20, and W, while the PI_tot_ differences were more profound since both B and R20-B80 had greater values compared to the rest of the treatments. Pak choi showed a different trend. Specifically, PI_abs_ was greater in R compared to W, while PI_tot_ was greater in B compared to W (Figure 1B,C).

Thet total soluble solids of lettuce were significantly greater in W compared to R, B, and R20-B80, where R was the second-best treatment while B showed the lowest values. In kale, the trend was similar to lettuce, but the values were significantly greater in W compared to all other treatments. Pak choi developed more total soluble solids under R compared to the rest of the treatments, R80-B20 and W followed, while B showed the lowest values (96% lower than R) (Figure 2A).

In lettuce, total phenolic content was significantly higher in R20-B80 compared to the rest of the treatments, followed by W, while R had the lowest values. Kale exhibited the greatest values under R20-B80 compared to all other treatments, with R also showing the lowest values (27% lower than R80-B20). In pak choi, total phenolic content was greater under R and R20-B80 compared to B and R80-B20 (Figure 2B).

The antioxidant activity displayed with the FRAP method showed similar trends with the total phenolic content. Specifically, lettuce showed greater antioxidant activity when grown under W or R20-B80 compared to all other treatments. Kale showed the greatest values under R20-B80 compared to the rest of the treatments. In pak choi, the total phenolic content was significantly greater under R20-B80 compared to B and R80-B20 (Figure 2C).

The total chlorophyll content of lettuce was significantly greater in R20-B80 compared to R, B, and R80-B20, while W was the second-best treatment and R and B had the lowest values (89–100% lower than R20-B80). A similar trend was observed in kale. Specifically, R20-B80 had the highest values compared to the rest of the treatments, with W being the second-best and B exhibiting the lowest (159% lower than R20-B80) values among all treatments. Conversely, in pak choi, B exhibited the greatest values (25–72% higher) compared to all other treatments, while R showed the lowest values among all treatments (Figure 3A).

The carotenoid content showed similar trends with the total chlorophyll content. Specifically, lettuce developed more total carotenoids under R20-B80 and W compared to R, B, and R80-B20. Similarly, kale showed the greatest values under R20-B80 and W compared to R, B, and R80-B20, with B exhibiting the lowest values (166–169% lower than R20-B80 and W). Pak choi developed more carotenoids under B compared to the rest of the treatments, followed by R, while R80-B20 and R20-B80 yielded the lowest values (Figure 3B).

Nitrates in lettuce were significantly greater under R compared to the rest of the treatments, followed by W, while B and R20-B80 had the lowest values. Quite similarly, pak choi had greater nitrate content under R compared to all other treatments, with R80-B20 and W following, while B and R20-B80 had the lowest values (83 and 82% lower than R). Kale accumulated more nitrates under W compared to all other treatments, followed by R20-B80, while B resulted the lowest values (Figure 3C).

Table 1 depicts the answers of eight of the restaurant’s employees on questions related to their subjective PFAL evaluation. Q1, Q4, and Q5 were answered positively (Positive and Moderately Positive) by 100% of the employees. Q2 and Q3 showed various responses among the employees. In particular, Q2 showed relatively positive responses (Positive and Moderately Positive were 62.5%) with 12.5% Moderately Negative responses. In addition, Q3 had the most diverse set of responses with 25% Positive, 37.5% Moderate, and 37.5% Negative responses.

## 3. Discussion

Lettuce and pak choi followed a very similar trend regarding their LMA performance. Specifically, the treatments with the greatest red-light emissions (R, R80-B20, and W) imposed the accumulation of the highest yield in these crops. Pak choi was probably expected to perform similarly to kale since both species are cruciferous (Brassicaceae) while lettuce belongs to the Asteraceae family. A similarity among all tested species was the decelerated growth rate under the monochromatic B light. It is known that monochromatic blue light suppresses the leaf growth, limiting the capacity to absorb light and drive photosynthesis. However, a combination of red-blue wavelengths is usually beneficial for plant growth, leading to greater yields. For example, a study with red pak choi demonstrated that a 3:1 red/blue combination led to greater shoot fresh weight compared to other treatments, including white and white supplemented with red [11]. In addition, red light has consistently been reported to enhance the growth of lettuce and other leafy and non-leafy vegetables as the most effective light spectrum for plant growth and photosynthesis [12]. A group of proteins called phytochromes are responsible for absorbing red light, which in turn induces plant growth and development, including the photosynthetic mechanism and the starch accumulation of [13]. For example, two lettuce cultivars showed increasing shoot fresh weight when grown under an increasing proportion of red light, while blue light induced the opposite result [14]. On the other hand, kale grown under different LED spectra (white or different red/blue ratios, as well as various radiation durations) did not show significant shoot fresh mass differences [15]. In another study, three kale cultivars did not exhibit significant effects in their fresh biomass when grown under three LEDs (different blue peak) or a cool-white fluorescent fixture [16].

PI_abs_ and PI_tot_ are two performance indices of the photosynthetic apparatus. In particular, PI_abs_ depicts the energetic efficiency from photon absorption by PSII to reduction in intersystem electron acceptors. In addition, PI_tot_ depicts the energetic efficiency from photon absorption by PSII to a reduction in PSI end acceptors [17]. Monochromatic red light has been found to damage the PSII antenna by limiting the biosynthesis of proteins such as CP43 and CP47 [18], whereas even a very low intensity of blue light (e.g., 5 μmol m^−2^ s^−1^ in Arabidopsis) restores the damage in a short amount of time [19].

From the absolute PI_abs_ and PI_tot_ values, it is clear that the plants were healthy and that no particular stress was at hand. Nevertheless, both parameters displayed a clear species dependency since the performance of each baby-leaf crop was variable under the different light conditions. For example, PI_abs_ in pak choi was enhanced under R, but the opposite was evident in lettuce, where R had the lowest values. In addition, R80-B20 showed relatively low values, mainly in lettuce and kale, reaffirming that treatments containing high amounts of red light generally lead to diminished photosynthetic performance. In another study, romaine lettuce treated with monochromatic red light exhibited poorer photosynthetic performance displayed by photosynthetic rate, stomatal conductance, and transpiration rate, as well as poorer chlorophyll fluorescence displayed by Fv/Fm, PI_abs_, PI_tot_, and other parameters related to heat dissipation, compared to red-blue light-containing treatments [20]. On the contrary, B and R20-B80 exhibited relatively high values in all crops, proving the necessity of both red and blue wavelengths for the proper function of the photosynthetic apparatus, particularly with emphasis on the blue part of the light spectra. The study of Hosseini et al. [21] involving green and purple basil showed the inhibition of PI_abs_ and other parameters related to the electron transient within the photosynthetic mechanism (i.e., RC/ABS, φ_P0_, and ψ_E0_) under the effect of red light compared to blue, white, and red-blue combinations.

The determination of total soluble solids is a means of quantifying the taste of vegetables. Higher total soluble solids are mainly composed of soluble sugars; thus, they are usually associated with greater approval from consumers, which also affects their purchasing behavior [22]. In all species, the lowest total soluble solids were detected under the most blue-containing treatments, B (the lowest) followed by R20-B80. In lettuce and pak choi, R resulted in significantly higher total soluble solids compared to B. Similarly to our findings, Chen et al. [23] also reported significantly lower soluble sugar content in lettuce, particularly fructose, glucose, sucrose, as well as a lower total sweetness index under monochromatic blue and even red-blue treatments compared to monochromatic red which led to higher contents and sweetness. The authors attributed their results to the enhanced activity of sucrose-degrading enzymes (acid invertase and neutral invertase) and the lower activity of sucrose synthesizing enzyme under monochromatic red.

Phenolic compounds are molecules directly associated with plant antioxidant activity [24]. The total phenolic compounds and antioxidant potential displayed by FRAP assay were enhanced in all tested species under the same light treatment, R20-B80. It is clear that a red-blue light environment rich in the blue spectral zone is beneficial for the production and accumulation of antioxidant compounds. Moreover, monochromatic B and treatments containing the most red, R and R80-B20, led to the lowest values in all species (except for R in pak choi). Other studies involving lettuce and kale reported similar findings. For example, Zhang et al. [25] found greater total phenolic content (anthocyanins, kaempferol, and quercetin) and higher antioxidant activity (DPPH radical scavenging rate and FRAP) in Chinese kale treated under a red-blue light with increased blue amount. In a study with red lettuce, a white-blue light induced greater total phenolic content, including certain phenolics (gallic acid, cyanidin, chlorogenic acid, vanillic acid, caffeic acid, and quercetin-3-O-glucopyranoside), as well as greater antioxidant activity (DPPH radical scavenging) compared to plain white, white-red, and white-green light environments [26]. The authors attributed the increase in most phenolics to the enhancement of phenylalanine ammonia-lyase (an important enzyme in the phenylpropanoid pathway) activity under the effect of a blue-rich environment, while the increase in cyaniding was attributed to the activation of important enzymes (dihydroflavonol-4-reductase and chalcone synthase) by blue light in the biosynthetic pathways of phenolic compounds [26,27].

The pigments related to the photosynthetic performance, chlorophylls and carotenoids, were consistent with the results of antioxidant activity. It is well-established that chlorophylls are essential molecules that capture photons mainly of blue and red wavelengths and use them to drive photosynthesis [28]. Moreover, carotenoids are light-harvesting molecules which also protect the photosynthetic apparatus from photodamage [29]. In our study, lettuce and kale were richer in both pigments under the effect of R20-B80, while pak choi exhibited more pigments under B. The treatments with the highest red light content, R and R80-B20, showed low pigment values in all species. Blue light acting through phototropin photoreceptors has been reported as a key wavelength for chlorophyll and carotenoid biosynthesis, which are synthesized to ameliorate tissue damage from excessive radiation [30]. Similarly, Fan et al. [31] also reported greater total chlorophyll content in pak choi grown under blue compared to white and red light, while the latter showed the lowest values. In a similar manner, red leaf lettuce grown for 17 days under blue-containing LEDs (blue or red-blue) formed greater amounts of chlorophyll and carotenoid molecules compared to monochromatic red [32], while in another study, red leaf lettuce also showed greater chlorophyll content under the most blue-containing treatments [14].

Nitrates are nitrogen-containing compounds that are associated with food safety and have a considerable presence in leafy vegetables. Nitrates have been associated with diseases such as gastric cancer and methemoglobinemia [33]. According to the European Commission Regulation No. 1258/2011 [34], nitrate concentration in lettuce must not exceed 2–5 g/kg of fresh weight, depending on the type of lettuce and season of harvest. The same EU regulation applies for spinach (3.5 g/kg of fresh weight) and rocket (6–7 g/kg of fresh weight) among leafy vegetables. In our case, the maximum mean nitrate values in lettuce were 0.66 g/kg, well within the limits set by EU. In addition, the maximum mean nitrate values were 1.00 g/kg in kale and 1.45 g/kg in pak choi, which were considerably lower compared to the EU limits set for another cruciferous leafy vegetable, rocket. The least nitrate content was found under monochromatic B in the three tested species. On the contrary, monochromatic R led to the highest amounts in lettuce and pak choi. This is in accordance with the findings reported by Fan et al. [31] in pak choi, where blue light enhanced the production of soluble protein and the accumulation of NADPH and enzymes, which were subsequently used for the reduction in nitrate. The authors attributed the lower nitrate content to the greater nitrate reductase, nitrite reductase, glutamine synthetase, and glutamate synthase activities measured in their study. Similarly to our findings, in another study including kale microgreens, Brazaitytė et al. [7] reported an increase in nitrate content under a 90–10% red-blue combination compared to monochromatic red or blue and other red-blue combinations. In another study involving spinach cultivated in an indoor vertical system, increasing red light also led to greater nitrate content [35].

The employees expressed an overall positive opinion about the presence of the PFAL within the restaurant compounds. From personal communication, it was determined that W treatment was the most desirable among the light treatments. This is not surprising since a color rendering index of over 50 units is favorable for the human eye, while it facilitates cultural practices such as scouting, harvest, etc. However, the employees did not express any negative responses regarding the other light treatments since they only spend short amounts of time with the PFAL and the crop production. Moreover, customers seem to express positive opinions towards the PFAL as a means to consume freshly produced vegetables with the lowest possible mileage.

## 4. Materials and Methods

### 4.1. Plant Material and Cultivation

The experiment was conducted in a PFAL located in the city center of Thessaloniki, Greece. The PFAL system was established on the first floor of a restaurant, within the cooking spaces, and can actually be visited (Ptolemeon 26, Thessaloniki, Greece; N 40.638; E 22.939).

Seeds of lettuce (*Lactuca sativa*) Green Battavia, kale (*Brassica oleracea* var. *sabellica*), and white pak choi (*Brassica rapa* var. *chinensis*) were bought by MP Seeds (Lodz, Poland). The seeds were sown in polystyrene trays (1.149 seedlings/m^2^; 300 × 435 mm), with two seeds per cell to ensure full tray coverage, and later thinned to one plant per cell. The cells were filled with peat substrate and regularly sprayed with water to ensure fine moisture conditions. Upon germination in dark conditions and 23 ± 2 and 15 ± 2 °C day/night temperatures, the trays were moved into polystyrene containers (320 × 470 mm), which marked the start of the experiment. The containers were filled with 16 L Hoagland nutrient solution (pH 5.6; electric conductivity 2.6 mS cm^−1^) [36], and the trays floated until the end of the experiment (25 days). One container was used per species and per light treatment and was frequently rotated on the shelf. After 15 days, the nutrient solution was substituted with a fresh one. The nutrient solution comprised the following nutrient concentrations (in ppm): N, 210; P, 31; K, 235; Ca, 200; Mg, 48; S, 64; Fe, 1; B, 0.11; Mn, 0.11; Zn, 0.023; Cu, 0.014; and Mo, 0.018. Day and night temperatures inside the room were about 23 ± 2 and 15 ± 2 °C, respectively, and relative humidity was 70 ± 10%; the room was ventilated, and the air recirculated sufficiently.

### 4.2. Light Conditions

Immediately after the experiment started, the containers were placed on five shelves (0.4 × 1.2 m) stacked vertically, corresponding to the five light treatments tested. Each shelf was illuminated by a set of LEDs (ZSP Technology Co., Ltd., Zhengzhou, China) emitting photosynthetic photon flux density of 180 ± 10 μmol m^−2^ s^−1^ (at the plant top) with 16 h photoperiod, but with different spectra. Specifically, the light treatments were 100% red (R; peak wavelength at 665 nm), 100% blue (B; peak wavelength at 449 nm), two red/blue combinations at 80/20% (R80-B20) and 20/80% (R20-B80), and a broad-spectrum white (W) light. These light treatments were previously tested in various combinations but under fully controlled conditions and were mostly designed for large-scale plant factories. In the present study, we tested the feasibility and efficiency of such light treatments in a PFAL established in a restaurant where several employees spend many hours within the spaces lit by the tested LEDs. Environmental conditions (i.e., temperature, relative humidity, and light) were uniform on each shelf. Parameters related to light conditions were determined with an HD 30.1 spectroradiometer (DeltaOhm Srl, Padova, Italy) and are presented in Table 2. Briefly, yield photon flux density (YPFD) is the total photon flux in the photosynthetically active radiation (PAR) region contributing to the plants’ yield. Phytochrome photostationary state (PPS) is an indicator of the balance between the Pr and Pfr forms of phytochrome, both of which participate in photomorphogenesis. High PPS values mean that more phytochrome is present in the Pfr form; thus, growth and development are expected to be promoted. Low PPS values mean that more phytochrome is present in the Pr form; thus, secondary metabolism might be triggered. The color rendering index (CRI) is an estimation of a light source’s ability to reflect the color of objects in a similar way to natural light. In plant photobiology, CRI does not provide information about the light source’s efficiency to impose growth and development. However, it is an index of the environment’s suitability for the human eye in terms of light quality.

### 4.3. Determinations

At 25 days post sowing and when the leaves reached a commercial size of about 10 cm in length (baby-leaf stage), photosynthetic performance indices (PI_abs_ and PI_tot_) were determined with a Pocket-PEA fluorometer (Pocket-PEA, Hansatech Instruments, Norflock, UK) on four plants per light treatment and after 20 min of dark adaptation.

Afterwards, the baby leaves were harvested 1 cm above the substrate, excluding the outer rows and columns of each growing tray, to eliminate leaves with possible light contamination from neighboring light sources. The remaining leaves were weighed to determine the leaf mass per area (LMA) and stored in a freezer (−20 °C) for phytochemical evaluations. LMA was calculated as follows: LMA (kg/m^2^) = FW (kg)/tray area (m^2^). (FW: total fresh weight of each tray. Tray area: the available planting space within a polystyrene tray after excluding the outer rows and columns, as mentioned above.)

The following day, the leaves were homogenized to obtain samples for the determination of total soluble solids (measured using a PAL-α refractometer, Atago, Tokyo, Japan), as well as the spectrophotometric quantification of other nutritional and developmental compounds (using a UV–Vis Spectrophotometer, Shimadzu, Kyoto, Japan).

Specifically, total phenolic content was measured according to Singleton and Rossi [37]. Briefly, leaves of each species were homogenized, and samples were added in falcon tubes with 80% methanol for phenolic extraction. Upon filtration, 2.5 mL 10% Folin–Ciocalteu reagent was added in each sample, followed by 2 mL 7.5% Na_2_CO_3_. Then, samples were incubated at 50 °C for 5 min, and absorbance of the colored reaction was measured at 760 nm versus a blank consisting of 80% methanol. Results were expressed as mg of gallic acid equivalent/g fresh weight.

The methodology of Benzie and Strain [38] for the determination of ferric-reducing antioxidant power (FRAP) was used to quantify total antioxidants. The same extract solution (80% methanol) was used during sample preparation. Briefly, 3 mL of a working solution (FeCl_3_, TPTZ, and CH_3_COONa buffer solution with pH 3.6) was added to each sample. Then, samples were incubated at 37 °C for 4 min, and the absorbance of the colored reaction was measured at 593 nm versus that of a blank consisting of 80% methanol. Results were expressed as μg of ascorbic acid equivalent/g fresh weight.

Total chlorophyll content and total carotenoid content were determined according to Sumanta et al. [39]. Briefly, pigments from homogenized leaves were extracted with 80% acetone. The samples were centrifuged at 10,000 rpm, 4 °C for 10 min and the colored extracts were added in 100 mL volumetric flasks. Absorbance of the colored reaction was measured at 663, 647, and 470 nm versus a blank consisting of 80% acetone. Pigment concentrations were calculated according to the following equations:$$Chl\;a=12.25*A_{663.2}-279*A_{646.8}$$
$$Chl\;b=21.50*A_{646.8}-5.10*A_{663.2}$$
$$Carotenoids=(1000*A_{470}-1.82*Chl\;a-85.02*Chl\;b)/198$$

Finally, nitrate content was determined according to the methodology described by Cataldo et al. [40]. Briefly, homogenized samples were added in falcon tubes with 25 mL water. An aliquot of 0.8 mL of H_2_SO_4_ and 5% salicylic acid was added to the 0.2 mL extract, followed by 19 mL of 2N NaOH. The absorbance of the colored reaction was measured at 410 nm. Results were expressed as mg/kg fresh weight.

### 4.4. Questionnaire

After the experiment ended, a questionnaire was handed to eight restaurant employees. The following questions were answered on a scale ranging from Negative to Positive (Negative—Moderately Negative—Moderate—Moderately Positive—Positive):Q1. How do you feel about being part of the vegetable growing process?Q2. Do you think that the cultural practices limit your time to fulfill other work-related obligations?Q3. After this experience, would you consider growing your own vegetables outside of work?Q4. What is the customers’ reaction upon visiting the PFAL?Q5. What is your overall evaluation of the PFAL within the restaurant compounds?

### 4.5. Statistical Analysis

The analysis of variances was performed using the SPSS package (SPSS 28.0, IBM Corp., Armonk, NY, USA). Each species was tested independently. A post hoc analysis of the means was conducted with Tukey’s HSD test at a significance level of α = 0.05.

The normality of the questionnaire responses was tested with the Kolmogorov–Smirnov test at a significance level of α = 0.05.

## 5. Conclusions

As a general rule, species dependency ensued in almost all tested parameters. Monochromatic blue led to the lowest LMA, total soluble solids, antioxidant activity, and nitrate content in all vegetables tested, as well as the lowest chlorophyll and carotenoid contents in lettuce and kale. Conversely, R20-B80 showed the greatest values in nutritional parameters, such as antioxidant activity, as well as phenolic, chlorophyll, and carotenoid contents. In addition, monochromatic red showed the highest LMA and nitrate content in lettuce and pak choi, as well as the greatest total soluble solids and phenolics in pak choi. The performance of the photosynthetic apparatus (displayed by PI_abs_ and PI_tot_) was diminished under increased red light with the exception of PI_abs_ in pak choi, which was enhanced under R. In most cases, R80-B20 showed less intense responses compared to red. In addition, white light led to greater LMA in kale and pak choi, FRAP in lettuce, carotenoids in lettuce and kale, and higher total soluble solids and nitrates in kale. The employees formed an overall positive opinion about the PFAL established in the restaurant compound, especially when white light was used, while customers also expressed positive feelings. Overall, our results on product quality are in agreement with other literature reports, even though our study was conducted in an environment where plant production is a secondary activity instead being part of a fully controlled research-oriented plant factory. It should be noted that the PFAL fulfills a large part of the restaurant’s needs for kale and pak choi but only a portion of the needs for lettuce baby leaves. Future research might shed light on the economic implications of such systems in terms of customer attraction, as well as production costs versus the wholesale market of leafy vegetables.

## Figures and Tables

**Figure 1 plants-14-00153-f001:**
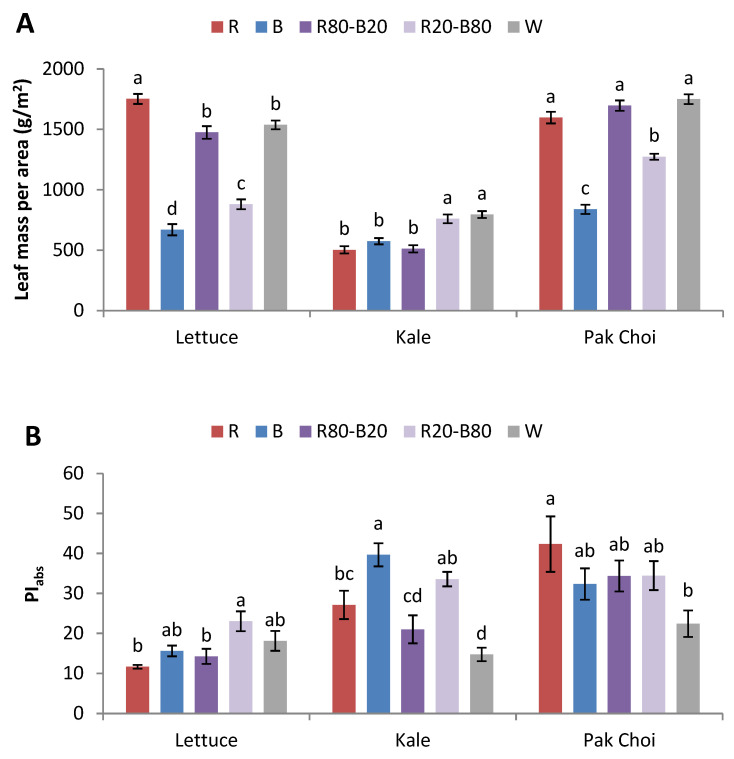
(**A**) Leaf mass per area and (**B**) PI_abs_ and (**C**) PI_tot_ of lettuce, kale, and pak choi baby leaves grown in a plant factory under five light treatments. Within each species, bars (±SE) followed by different letters are significantly different (*p* ≤ 0.05).

**Figure 2 plants-14-00153-f002:**
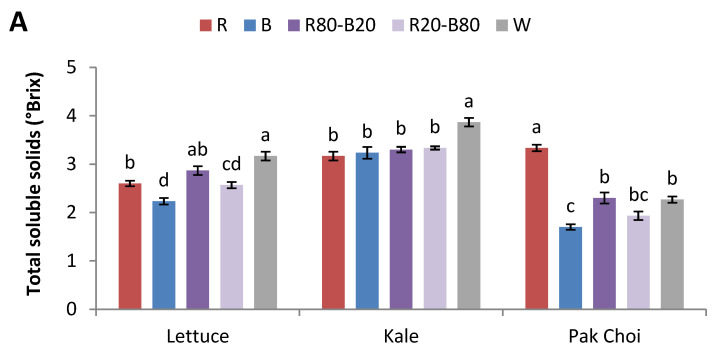
(**A**) Total soluble solids, (**B**) total phenolic content, and (**C**) antioxidant activity (FRAP) of lettuce, kale, and pak choi baby leaves grown in a plant factory under five light treatments. Within each species, bars (±SE) followed by different letters are significantly different (*p* ≤ 0.05).

**Figure 3 plants-14-00153-f003:**
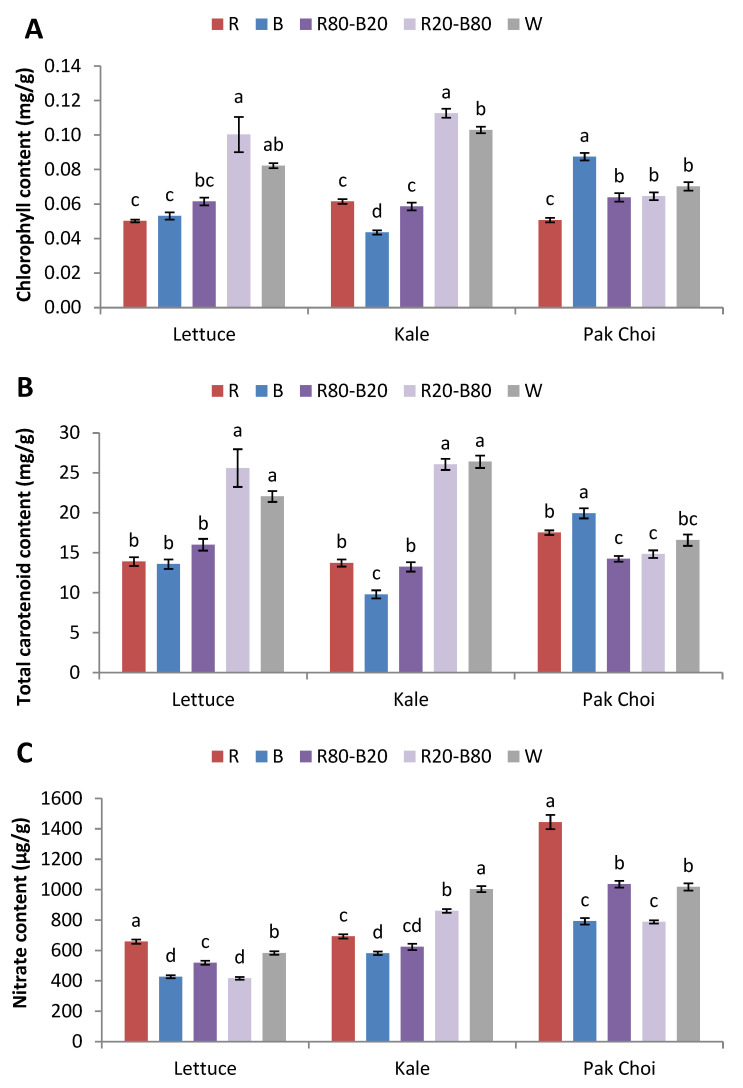
(**A**) Total chlorophyll content, (**B**) total carotenoid content, and (**C**) nitrate content of lettuce, kale, and pak choi baby leaves grown in a plant factory under five light treatments. Within each species, bars (±SE) followed by different letters are significantly different (*p* ≤ 0.05).

**Table 1 plants-14-00153-t001:** Set of questions answered by eight employees of the restaurant. The answers ranged from Negative to Positive (Negative—Moderately Negative—Moderate—Moderately Positive—Positive): Q1. How do you feel about being part of the vegetable growing process? Q2. Do you think that the cultural practices limit your time to fulfill other work-related obligations? Q3. After this experience, would you consider growing your own vegetables outside of work? Q4. What is the customers’ reaction upon visiting the PFAL? Q5. What is your overall evaluation on the PFAL within the restaurant compounds?

Answer	Q1	Q2	Q3	Q4	Q5
Positive %	87.5	50	25	50	50
Moderately Positive %	12.5	12.5	0	50	50
Moderate %	0	25	37.5	0	0
Moderately Negative %	0	12.5	0	0	0
Negative %	0	0	37.5	0	0
Normality (sign. two-tailed)	<0.001	0.034	0.200	0.001	0.001

**Table 2 plants-14-00153-t002:** Wavelength distribution, yield photon flux density (YPFD), phytochrome photostationary state (PPS), and color rendering index (CRI) of the light treatments tested. YPFD, PPS, and CRI were calculated according to Sager and McFarlane [28].

Waveband	Light Treatment				
	R	B	R80-B20	R20-B80	W
UV %; 380–399 nm	0	0	0	0	0
Blue %; 400–499 nm	0	100	20	80	20
Green %; 500–599 nm	0	0	0	0	22
Red %; 600–699 nm	100	0	80	20	48
Far-red %; 700–780 nm	0	0	0	0	10
YPFD (μmol m^−2^ s^−1^)	79.1	63.8	89.2	77.8	80.8
PPS	0.89	0.51	0.89	0.83	0.81
CRI	−95.0	−74.8	−152.7	−70.0	58.5

## Data Availability

Data are contained within the article.

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
