# Peer review of "Plant Factory in a Restaurant: Light Quality Effects on the Development, Physiology, and Quality of Three Baby-Leaf Vegetables"

_plants, 2025, doi:10.3390/plants14020153_

Round 1
Reviewer 1 Report
Comments and Suggestions for Authors
The manuscript deals with a very popular topic of indoor farming. The manuscript is written clearly and well-structured. However, I have some concerns about the suitability of this manuscript to the journal and the content, which is why I suggest rejection. I reached this conclusion after reading the aims and scope of the journal and the special issue. I recommend resubmitting the manuscript to another journal.
I also have some comments on the content. Below, the Authors can find the main ones:
1. I recommend removing the abbreviation from the title of the manuscript.
2. No research gap is underlined in the literature review – The authors suggest the lack of information on the effect of light on the growth of kale and pak choi and put it as the main objective. In the literature, I found reports on the effect of spectral composition on kale and pak choi growth, which makes this aim groundless. Please clarify the research gap.
3. The results reported in Table 1 are non-significant. I do not think reporting acceptability based on answers from 8 people brings any reliable information.
4. How were the moisture conditions controlled?
5. What is PPS, and what is its role in the evaluation of lighting for indoor farming?
6. What is the justification for providing information on CRI?
Author Response
Comment: The manuscript deals with a very popular topic of indoor farming. The manuscript is written clearly and well-structured. However, I have some concerns about the suitability of this manuscript to the journal and the content, which is why I suggest rejection. I reached this conclusion after reading the aims and scope of the journal and the special issue. I recommend resubmitting the manuscript to another journal.
Response: The authors would like to express their gratitude to the reviewers for the time they invested for assessing our manuscript and their useful comments and suggestions. Regarding the reviewer’s concerns about the suitability of the manuscript for this journal, we would like to stress out that our manuscript was invited from “Plants”, while its suitability was already checked by the editors before submission. The comments were responded one-by-one below.
Comment: I also have some comments on the content. Below, the Authors can find the main ones:
- I recommend removing the abbreviation from the title of the manuscript.
Response: The term “PFAL” was substituted by “Plant factory” in the title as suggested.
Comment:
- No research gap is underlined in the literature review – The authors suggest the lack of information on the effect of light on the growth of kale and pak choi and put it as the main objective. In the literature, I found reports on the effect of spectral composition on kale and pak choi growth, which makes this aim groundless. Please clarify the research gap.
Response: The objectives were amended accordingly; please check L79-100. Contrary to most research studies conducted in growth rooms under fully controlled conditions, our research focuses on the potential baby-leaf cultivation in the busy spaces of a restaurant. Therefore, we did not aim to simulate the cultural conditions of a production unit, but to test the cultivation of leafy vegetables close to the actual consumer’s plate, in the final stage of the “farm-to-fork” chain. An important aim was to test whether the traditionally used light treatments are suitable for such establishments with respect to the obtained product quality, as well as the practicality for employees and the acceptance from customers (secondarily). The study provides us with the opportunity to evaluate the real-life challenges and benefits ensuing during actual operation of a restaurant.
Comment:
- The results reported in Table 1 are non-significant. I do not think reporting acceptability based on answers from 8 people brings any reliable information.
Response: To our knowledge this is the first article providing research findings of a plant factory established in a restaurant instead of a fully-controlled growth room, including the opinion of employees who are not fully trained on vegetable production. The vast majority of research studies related to plant factories are conducted in rooms with fully controlled conditions, and targeted for similar but large-scale commercial establishments which focus solely on crop production. In such establishments, employees are typically trained for crop production. However, plant factories are nowadays established in hotels, restaurants and other commercial spaces where employees are usually untrained to grow vegetables. The questionnaire aimed to capture the opinion of such employees taking into consideration that vegetable growing is not their main obligation during their shift (Q1, Q2, and Q3). Q4 refers to the customers’ acceptance of the PFAL, possibly leading to a greater turnover for the restaurant. However, an economic analysis have not yet been conducted. It is one of our near future objectives. Moreover, the PFAL operates within the cooking spaces, a place where employees spend a lot of hours. We hypothesized that several light spectra (blue, red, and their combinations) would probably be undesired by the employees due to the low CRI and the color distortion of various objects and equipment in the room. However, this was not the case since the employees answered positively in Q5, as well as in personal communication.
Comment:
- How were the moisture conditions controlled?
Response: As mentioned in the manuscript the PFAL was established in the cooking spaces of a restaurant. The relative humidity was monitored at 70 ± 10% and was partly controlled through ventilation and air recirculation with fans (L351-352).
Comment:
- What is PPS, and what is its role in the evaluation of lighting for indoor farming?
Response: PPS (phytochrome photostationary state) is an indicator of the balance between the Pr and Pfr forms of phytochrome, both of which participate in photomorphogenesis. PPS provides information related to plant growth and development. High PPS values (e.g. greater than 0.80) mean that more phytochrome is in Pfr form, thus growth and development are expected to be promoted. Low PPS values (e.g. lower than 0.60) mean that more phytochrome is in Pr form, thus secondary metabolism might be triggered. The manuscript was enriched with information in L371-376.
Comment:
- What is the justification for providing information on CRI?
Response: CRI (color rendering index) is an estimation of a light source’s ability to reflect the color of objects in a similar way with natural light. In plant photobiology, CRI does not provide information about the light source’s efficiency to impose growth and development. However, it is an index of the environment’s suitability for the human eye in terms of light quality. CRI was already mentioned in the Discussion, in L319-323. However, the manuscript was also enriched with information in L376-380.
Reviewer 2 Report
Comments and Suggestions for Authors
The manuscript fits perfectly into the scope of the journal. The authors have taken up the interesting topic of urban agricultures. The topic is very interesting and fits into the general trends of research related to climate change. The authors investigated the PFAL system with different lighting conditions for the cultivation of three plant species used as culinary plants. The manuscript may be published after corrections. Below are my suggestions for corrections that might improve the manuscript.
In the Introduction section, I propose to provide the Latin names of the plants studied and their family affiliation.
In line 89-90 the explanation "B: monochromatic B" is probably wrong
Line 92 "leaf mass per area" - the methodology does not discuss how the LMA measurements and calculations were performed.
Line 251 "cyaniding" - why was it listed among phenolic compounds?
Lines 357 - 366 - the methodology is too brief. Please describe the research methodology, the method of extract preparation and the parameters of the assays.
In the Conclusion section, please refer to the economic aspect of the cultures studied.
Comments on the Quality of English LanguageI would like to ask for linguistic proofreading of the manuscript because some of the wording is unclear.
Author Response
Comment: The manuscript fits perfectly into the scope of the journal. The authors have taken up the interesting topic of urban agricultures. The topic is very interesting and fits into the general trends of research related to climate change. The authors investigated the PFAL system with different lighting conditions for the cultivation of three plant species used as culinary plants. The manuscript may be published after corrections. Below are my suggestions for corrections that might improve the manuscript.
Response: The authors would like to express their gratitude to the reviewers for the time they invested for assessing our manuscript and their useful comments and suggestions. The comments were responded one-by-one below.
Comment: In the Introduction section, I propose to provide the Latin names of the plants studied and their family affiliation.
Response, L85-87: The latin names and families were included in the manuscript as suggested.
Comment: In line 89-90 the explanation "B: monochromatic B" is probably wrong
Response, L102-103: Indeed, the term was corrected to “B: monochromatic blue”.
Comment: Line 92 "leaf mass per area" - the methodology does not discuss how the LMA measurements and calculations were performed.
Response, L395-397: LMA was calculated as follows: LMA (kg/m2) = FW (kg) / Tray area (m2). FW: total fresh weight of each tray. Tray area: the available plant space within a polystyrene tray. It is now included in the manuscript.
Comment: Line 251 "cyaniding" - why was it listed among phenolic compounds?
Response, L271: Indeed, the correct term is “cyanidin”. Cyanidin is a type of flavonoids (anthocyanin). It is now corrected in the manuscript. Thank you for the observation.
Comment: Lines 357 - 366 - the methodology is too brief. Please describe the research methodology, the method of extract preparation and the parameters of the assays.
Response, L402-426: The research methodology was enriched as suggested.
Comment: In the Conclusion section, please refer to the economic aspect of the cultures studied.
Response: According to the restaurant owners, the PFAL fulfills a large part of the restaurant’s needs for kale and pak choi but only a portion of the needs for lettuce baby leaves. It is now included in the conclusions, in L466-467. However, an economic analysis has not been conducted yet. It is one of our near-future objectives.
Comment: I would like to ask for linguistic proofreading of the manuscript because some of the wording is unclear.
Response: Regarding the English language, the manuscript was reviewed by Professor Kalliopi Radoglou, an esteemed colleague of ours with extensive experience. The manuscript was amended according to her suggestions.
Reviewer 3 Report
Comments and Suggestions for Authors
The manuscript describes a study of the effect of light quality on the growth of baby-leaf vegetables. The manuscript requires serious revision for several reasons:
1. The main drawback of the study is the lack of novelty and weak justification for the relevance of this study. Currently, there are many published studies on the effect of blue and red light and their combination on lettuce, pak choi and other cruciferous plants. The authors should emphasize the novelty of the study compared to the previous ones. The results obtained by the authors seem obvious, for example, the effect of slowing down the growth rate under monochromatic blue light is well known. It is also known that blue light stimulates the accumulation of phenolic compounds and chlorophylls. The article does not provide new knowledge about the effect of light quality on the studied plants. The mechanisms and causes of the results obtained are also not disclosed.
2. In the materials and methods section, justify the choice of light treatment modes. Similar lighting modes have already been used on cruciferous crops.
3. Line 218-220 "treatments containing high amounts of red light generally lead to diminished photosynthetic performance" requires clarification and references.
4. It is unclear for what purpose the staff was surveyed regarding the creation and operation of such a system within the restaurant complexes.
Author Response
Comment: The manuscript describes a study of the effect of light quality on the growth of baby-leaf vegetables. The manuscript requires serious revision for several reasons:
Response: The authors would like to express their gratitude to the reviewers for the time they invested for assessing our manuscript and their useful comments and suggestions. The comments were responded one-by-one below.
Comment:
- The main drawback of the study is the lack of novelty and weak justification for the relevance of this study. Currently, there are many published studies on the effect of blue and red light and their combination on lettuce, pak choi and other cruciferous plants. The authors should emphasize the novelty of the study compared to the previous ones. The results obtained by the authors seem obvious, for example, the effect of slowing down the growth rate under monochromatic blue light is well known. It is also known that blue light stimulates the accumulation of phenolic compounds and chlorophylls. The article does not provide new knowledge about the effect of light quality on the studied plants. The mechanisms and causes of the results obtained are also not disclosed.
Response: The objectives and conclusions were amended accordingly; please check L79-100 and L463-467. Contrary to most research studies conducted in growth rooms under fully controlled conditions, our research focuses on the potential baby-leaf cultivation in the busy spaces of a restaurant. Therefore, we did not aim to simulate the cultural conditions of a production unit, but to test the cultivation of leafy vegetables close to the actual consumer’s plate, in the final stage of the “farm-to-fork” chain. An important aim was to test whether the traditionally used light treatments are suitable for such establishments with respect to the obtained product quality, as well as the practicality for employees and the acceptance from customers (secondarily). The study provides us with the opportunity to evaluate the real-life challenges and benefits ensuing during actual operation of a restaurant. Moreover,
Comment:
- In the materials and methods section, justify the choice of light treatment modes. Similar lighting modes have already been used on cruciferous crops.
Response: Explanation about our choice of light treatments was included in L362-367.
Comment:
- Line 218-220 "treatments containing high amounts of red light generally lead to diminished photosynthetic performance" requires clarification and references.
Response, L233-246: A Literature finding was included in this paragraph. Also, the article of Hosseini et al. (2019) (already mentioned in this paragraph) reaffirms this statement.
Comment:
- It is unclear for what purpose the staff was surveyed regarding the creation and operation of such a system within the restaurant complexes.
Response: To our knowledge this is the first article providing research findings of a plant factory established in a restaurant instead of a fully-controlled growth room, including the opinion of employees who are not fully trained on vegetable production. The vast majority of research studies related to plant factories are conducted in rooms with fully controlled conditions, and targeted for similar but large-scale establishments. In such establishments, employees are typically trained for crop production. However, plant factories are nowadays established in hotels, restaurants and other commercial spaces where employees are usually untrained to grow vegetables. The questionnaire aimed to capture the opinion of such employees taking into consideration that vegetable growing is not their main obligation during their shift (Q1, Q2, and Q3). Q4 refers to the customers’ acceptance of the PFAL, possibly leading to a greater turnover for the restaurant. However, an economic analysis have not yet been conducted. It is one of our near future objectives. Moreover, the PFAL operates within the cooking spaces, a place where employees spend a lot of hours. We hypothesized that several light spectra (blue, red, and their combinations) would probably be undesired by the employees due to the low CRI and the color distortion of various objects and equipment in the room. However, this was not the case since the employees answered positively in Q5, as well as in personal communication.
Round 2
Reviewer 1 Report
Comments and Suggestions for Authors
Thank you for the clarifications. In my opinion they improve the manuscript enough to be published if Editors find it suitable for the journal.
Reviewer 3 Report
Comments and Suggestions for Authors
The comments have been corrected and the article has been supplemented. I have no questions.